# Inorganic Nanoflowers—Synthetic Strategies and Physicochemical Properties for Biomedical Applications: A Review

**DOI:** 10.3390/pharmaceutics14091887

**Published:** 2022-09-06

**Authors:** Su Jung Lee, Hongje Jang, Do Nam Lee

**Affiliations:** 1Ingenium College of Liberal Arts (Chemistry), Kwangwoon University, Seoul 01897, Korea; 2Department of Chemistry, Kwangwoon University, Seoul 01897, Korea

**Keywords:** inorganic nanoflowers, biomedical application, drug delivery, wound healing, antibacterial treatment, combinatorial cancer therapy, amyloidosis, H_2_O_2_, biosensors

## Abstract

Nanoflowers, which are flower-shaped nanomaterials, have attracted significant attention from scientists due to their unique morphologies, facile synthetic methods, and physicochemical properties such as a high surface-to-volume ratio, enhanced charge transfer and carrier immobility, and an increased surface reaction efficiency. Nanoflowers can be synthesized using inorganic or organic materials, or a combination of both (called a hybrid), and are mainly used for biomedical applications. Thus far, researchers have focused on hybrid nanoflowers and only a few studies on inorganic nanoflowers have been reported. For the first time in the literature, we have consolidated all the reports on the biomedical applications of inorganic nanoflowers in this review. Herein, we review some important inorganic nanoflowers, which have applications in antibacterial treatment, wound healing, combinatorial cancer therapy, drug delivery, and biosensors to detect diseased conditions such as diabetes, amyloidosis, and hydrogen peroxide poisoning. In addition, we discuss the recent advances in their biomedical applications and preparation methods. Finally, we provide a perspective on the current trends and potential future directions in nanoflower research. The development of inorganic nanoflowers for biomedical applications has been limited to date. Therefore, a diverse range of nanoflowers comprising inorganic elements and materials with composite structures must be synthesized using ecofriendly synthetic strategies.

## 1. Introduction

Flower-shaped nanomaterials called nanoflowers [1] have attracted the attention of researchers worldwide due to the multilayered structural characteristics of their petals. These nanoflowers have a higher surface-to-volume ratio than solid spherical nanoparticles, exhibit enhanced charge transfer and carrier immobility, and are highly efficient in surface reactions [2]. The syntheses and applications of nanoflowers have been widely investigated thus far. The flowerlike nanostructures are constructed using inorganic or organic materials, or a combination of both materials (called a hybrid), and are used in catalysts [3,4,5], dye-sensitized solar cells [6,7,8], lithium-ion batteries [9,10,11], supercapacitors [12,13], water splitting [14,15], and biomedical applications [16,17,18,19,20]. New materials with high therapeutic efficiencies that are inexpensive to synthesize with simple, robust, and eco-friendly synthesis routes are essential in biomedical science. Nanoflowers with branched structures satisfy all these requirements.

Nanoflowers can be classified on the basis of their composition: inorganic, organic, and hybrid (both organic and inorganic components); see Figure 1. Inorganic nanoflowers are composed of exclusively inorganic materials such as metals, metal oxides, alloys, and metalloids, or the inorganic materials are coated or doped using metalloids, carbon, nitride, sulfide, phosphide, selenide, and telluride [21,22,23,24,25,26,27,28,29,30,31,32]. Vesicles made from gemini amphiphiles that directed synthesis of Au nanoflowers were reported [33]. NiO nanoflowers are synthesized via a simple surfactant-free hydrothermal route employing Ni(NO_3_)_2_ and triethylamine followed by calcination. On the other hand, the NiO morphologies formed by synthesis with urea instead of triethylamine varied depending on the calcination temperature and produced nanoparticles or nanoslices at calcination temperatures of 400 and 600 °C, respectively [34]. Imura et al. introduced the preparation method of silica-coated Au nanoflowers on alumina to prevent the aggregation of the nanoflowers and precipitation [26]. Carbon-coated Fe_3_O_4_ nanoflowers were synthesized via a one-pot solvothermal route for biosensors in lateral flow immunoassays [27].

Organic nanoflowers are composed of organic molecules only or contain inorganic elements as part of the medium in which the organic molecules are the main components [35,36,37,38,39]. For instance, carbon nanoflowers synthesized using an electric arc discharge method in water were reported; the obtained carbon nanoflowers were composed of highly crystalline graphene nanosheets that were arranged like flowers [40]. Zheng’s group introduced nitrogen-, phosphorous-, and fluorine-doped carbon nanoflowers using ultrasound-induced polycondensation and pyrolysis [41]. In addition, organic nanoflowers have been constructed from a variety of molecules (guests) using fixed supramolecular hacky sacks, which are hierarchically structured particles, as templates [42]. Nanoflowers with spiky and wide petals have been produced by using small/rigid molecules (e.g., doxorubicin) and large/flexible biomacromolecules (e.g., proteins and plasmid DNA), respectively, as guests [42].

Organic–inorganic hybrid nanoflowers, also called hybrid nanoflowers, are defined as all components of inorganic nanostructures being associated with organic materials [43]. Generally, organic components include enzymes, proteins, amino acids, biopolymers, DNA, and peptides containing amide or amine groups to form complexes with metal ions via coordination interaction. Inorganic materials are mostly composed of divalent metals such as Cu^2+^, Zn^2+^, Ca^2+^, Fe^2+^, and Mn^2+^ [44]. Ge and coworkers [45] discovered the first hybrid nanoflowers, confirming that Cu^2+^ ions and proteins could be used to construct novel types of nanoparticles. The four types of hybrid nanoflowers were prepared using α-lactalbumin, laccase, carbonic anhydrase, and lipase, respectively. The formation of dual-enzyme inorganic hybrid nanoflowers was reported by using glucose oxidase and lipase as the organic materials and Cu_3_(PO_4_)_2_·3H_2_O as the inorganic components [46]. Li et al. synthesized carbon-nanotube-embedded lipase–Ca/Fe/Cu nanoflowers as a biocatalysts of the chiral resolution reaction [47].

Since the early 2000s, inorganic nanoflowers; i.e., nanoflowers synthesized from inorganic elements, have attracted the attention of researchers due to their unique nanostructural characteristics, as well as their excellent catalytic efficiency and optical properties, depending on their composition, crystal structure, and localized surface plasmon resonance (LSPR). In particular, inorganic nanoflowers have been widely used in photocatalysis applications such as plasmon-enhanced hydrogen evolution and alcohol oxidation [48,49]. The maximum absorption wavelength of LSPR in surface-enhanced Raman scattering (SERS) can be controlled by the nanoflower size for qualitative and quantitative analysis [50,51]. Nanoflowers are also widely used in energy applications and catalysis. Despite these advantages, fewer studies on inorganic nanoflowers for biomedical applications have been published than those on hybrid nanoflowers. In particular, the scope of inorganic materials used to synthesize nanoflowers requires expansion. To the best of our knowledge, this article is the first review of inorganic nanoflowers and their biomedical applications in the literature.

In this review, we will summarize the preparation, properties, and recent advances in inorganic nanoflowers in biomedical applications. First, we present some popular synthetic strategies for nanoflowers suitable for use in biomedical applications. Next, we provide a brief overview of the biomedical applications of the flowerlike nanostructures: their antibacterial effects in wound healing; medical devices and implants; biosensors to detect diseased conditions such as diabetes, food poisoning, and amyloidosis; drug delivery; and combinational treatment. Finally, we review the properties and efficiency of three-dimensional (3D) flower-shaped nanomaterials. Further, we discuss the current and future research trends in nanoflower research.

## 2. Synthesis and Characterization of Nanoflowers

In 2008, Xie et al. [52] reported on the three popular synthetic strategies of nanoflowers: soft-template-based synthesis (e.g., using liposomes as soft templates to guide the formation of flowerlike nanostructures) [53,54], anisotropic growth using capping agents such as poly (polyvinylpyrrolidone) (PVP) and cetyltrimethylammonium bromide (CTAB) [55,56,57,58], and oriented attachment of primary nanoparticles (e.g., synthesis of dendritic PtRu nanoparticles from faceted PtRu primary nanoparticles) [59]. Since then, various other methods have been applied to synthesize inorganic nanoflowers based on the previously reported nanomaterial synthesis technologies: physical, chemical, biological, and hybrid methods [60].

The first technology is the physical method, which is mainly represented by vapor technology. For instance, Bi_2_S_3_ nanoflowers grown on silicon substrate via a simple vapor deposition method were reported [61]. The morphology of the Bi_2_S_3_ nanostructure was controlled from flowers to bundles of nanorods by controlling the partial pressure of the reactant as the experimental parameter. The second synthesis method is the chemical synthesis strategy, which has been most widely applied to form inorganic nanoflowers. The colloidal [62], sol-gel [63], inverse micelles [64,65], hydrothermal [6,66], solvothermal [67,68,69], electrodeposition [70,71], and microwave synthesis [72,73] methods have been studied in the formation of inorganic nanoflowers. Mo_1−x_W_x_Se_2_ alloy nanomaterials with nanoflower morphologies were synthesized using the controlled colloidal synthesis of composition and morphology [62]. The similar morphologies were maintained while changing the composition. As the content of W(x→1) increased, the size and thickness of the sheet slightly increased. The Fe_3_O_4_@MnO_2_ core–shell nanoflowers were fabricated via a solvothermal method [68]. Further, the 3D ultrafine Pt nanoflower was directly deposited on the carbon-coated gas diffusion layer electrode by the electrodeposition method [70]. The microwave synthesis strategy was used to hierarchically structure NiCo_2_O_4_ nanoflowers [73]. Third, among other biological nanomaterial synthesis methods, the biological synthesis approach known as green synthesis that uses plant extracts (e.g., *Azadirachta indica* leaves [74], *Dodonaea angustifolia* [75], *Kalanchoe daigremontiana* [16], *Ocimum sanctum (Tulsi)* leaves [76], and *Withania coagulans* [77]) has been mainly studied. Bioinspired synthesis of ZnO nanoflowers was introduced using a *Withania coagulans* extract as the reducing agent [77]. The last example of synthesis technology for flower-shaped inorganic nanomaterials is the hybrid nanomaterial synthesis method, which is a multistep synthesis method that combines various physical, chemical, and biological methods such as electrochemical deposition [78,79], chemical vapor deposition [80], high-energy ball-milling hydrothermal treatment [81], and the solution-immersion RF-sputtering method [82]. MoSe_2_ nanoflowers on a 3D carbon cloth surface were fabricated using chemical vapor deposition by controlling the temperature and growth time in order to manipulate the morphology, thickness, and formation of both Mo and Se active edge sites [80]. Ag@NiO core–shell nanoflower arrays were prepared using the one-step solution-immersion process and subsequent RF-sputtering method [82].

Organic nanoflowers were synthesized using similar synthesis technologies to those of inorganic nanoflowers. The electric arc discharge method [40], ultrasound-induced polycondensation and pyrolysis [41], reduction–pyrolysis–catalysis route [83], chemical vapor deposition [84], and microwave-assisted high-temperature/hydrothermal carbonization etching method [85,86] were reported.

In the case of the organic–inorganic hybrid nanoflower first reported in 2012, the mild and direct coprecipitation method was developed to synthesize hybrid protein–copper phosphate nanoflowers [45]. Since then, various hybrid nanoflowers have been synthesized based on the coprecipitation method, which was carried out by mixing the organic elements (e.g., enzymes, protein, amino acids, and so on) and metal ions (Cu^2+^, Zn^2+^, Mn^2+^, Fe^2+^, and Co^2+^) in the presence of phosphate-buffered saline or directly using metal phosphate. The mixture was then incubated or sonicated [87,88,89,90,91].

Herein, we introduce some specific synthesis strategies, especially for biomedical applications.

Flowerlike nanomaterials with a hollow morphology, for drug delivery, can be synthesized using challenging template-based synthesis. For example, hollow Au nanoflowers (H–AuNFs) were synthesized using polyacrylic acid (PAA) nanospheres as templates (Figure 2a) [20]. Briefly, small Au nanoparticles were formed on a PAA nanosphere surface by the addition of chloroauric acid. The seed-mediated growth of the as-synthesized Au nanoparticles through the reduction by l-ascorbic acid led to the formation of the nanoflowers with a hollow morphology. After the reaction, PAA nanosphere templates were easily removed by washing with deionized water. As shown in Figure 2c,e, the H–AuNFs prepared using the template-mediated method exhibited a flowerlike and hollow morphology with a 450 nm diameter.

As shown in Figure 3, Cu_2_O nanocubes have been used as challenging templates for synthesizing hierarchical CuO nanoflowers [92]. The addition of H_2_O_2_ gradually oxidized the Cu_2_O nanocubes, resulting in the formation of ultrathin CuO nanosheets on their surface. As the reaction time was increased, the inner Cu_2_O nanocubes gradually disappeared and the nanosheets increased in size, thus generating hierarchical CuO nanoflowers with ultrathin nanosheets. As shown in Figure 3b,c, the CuO nanoflowers were composed of numerous crooked nanosheets with a thickness of <10 nm and a large surface area (78.35 m^2^ g^−1^). As shown in the HR-TEM image (Figure 3d), the measured lattice spacings of 0.234 nm and 0.236 nm were ascribed to the (111) plane of CuO. The XRD pattern and XPS spectra in Figure 3e–g indicate that the CuO had a monoclinic geometry [93,94].

To synthesize nanoflowers, PVP and CTAB were used as capping agents and surfactants for structural control, but they were difficult to remove from the surface of the nanomaterial, requiring severe conditions or multiple washings [95]. Nanoflowers can also be synthesized via anisotropic growth by using biocompatible Good’s buffers such as 3-[4-(2-hydroxyethyl)piperazin-1-yl]propane-1-sulfonic acid (EPPS) and 2-[4-(2-hydroxyethyl)-1-piperazinyl]ethanesulfonic acid (HEPES) as reducing and shape-directing agents [96]. HEPES has good biocompatibility and environmental and cost advantages, and also provides a clean surface where postsynthesis surface modifications can be easily performed for biological applications [52].

Size-controlled metallic Au nanocrystals with flowerlike structures were synthesized in high yields with excellent monodispersity via a modified HEPES reduction method without seeds or surfactants [52,95]. Increasing the HEPES concentration to 15 and 20 mM induced the formation of smaller nanoflowers with diameters of 65 ± 8 nm (Figure 4a,b) and 48 ± 6 nm, respectively (Figure 4c,d). A further increase in the HEPES concentration to 40 mM resulted in the formation of spherical and irregular-shaped nanoparticles with an approximately 5 ± 35 nm diameter (Figure 4e,f). After a reaction period of 8 min, the solid product consisted of primary Au nanocrystals with diameters of 2 ± 20 nm (Figure 5b). This reaction period of nanoflower formation; i.e., nucleation of the primary Au nanocrystals, was called Stage 1. These primary nanocrystals were unstable and agglomerated to reduce the overall surface energy for the next 4 min of reaction, which was the beginning of Stage 2. The morphology of the product was studied at 12 min when the agglomerates comprised tens of primary crystals, as shown in the TEM image (Figure 5b). The reduction rate decreased after a long period depending on the consumption of the Au precursor, which was the limiting reactant; at this point, Au was deposited in energetically favorable directions, causing anisotropic growth of the agglomerates. Flowerlike nanostructures grew from the branches protruding from the surfaces of the agglomerates, which was labeled Stage 3. TEM images showed the formation of highly branched Au nanostructures through the anisotropic growth of agglomerates from Stage 2 until 24 min of reaction time (Figure 5b).

Another approach to synthesizing nanoflowers involves a core nanoparticle and flower-shaped shells of different compositions encapsulating the surface of the core nanoparticle. For instance, highly bioactive and low-cytotoxic Si-based NiOOH nanoflowers were synthesized using a modified chemical bath deposition method [24,97]. As shown in Figure 6, plasma-synthesized silicon nanoparticles of a 50–100 nm particle size were encapsulated in porous flowerlike NiOOH shells so that the diameters of the Si-based nanoflowers were in the range of 500 nm–1 μm and thickness of the porous NiOOH shell layers was 200–450 nm (Figure 7a). As shown in Figure 7b, the highly intense diffraction peaks in the XRD patterns corresponded to Si and the peaks at 2θ = 12° and 24° were attributed to the NiOOH coated on the Si nanoparticles. The formation of the Si@NiOOH was also confirmed using UV–vis spectroscopy.

To put it shortly, inorganic nanoflowers are prepared in general using four types of synthesis technologies (e.g., physical, chemical, biological, and hybrid methods), based on previously reported synthesis strategies of nanomaterials. As the shape and particle size of nanoflowers and the thickness of the petals are influenced by the synthesis method, experimental parameters, composition ratio, and structures [34,62,77,98,99,100], the development of new synthesis methods is essential. Moreover, the specific synthesis strategies for biomedical applications have been described. The first example is a colloidal method (with the challenging template) of synthesizing hollow-shaped nanoflowers designed for drug transport. The next example is the seedless synthesis method using HEPES, a zwitter-ionic organic buffering agent that has minimal salt and temperature effects and high water solubility. The final example is the encapsulation of the surface of the core nanoparticle using different components. Various surface components can be applied that can be easily utilized in wider biomedical applications. Therefore, in order to develop biomedically applied nanoflowers, it is important to develop synthesis methods using reactants with high biocompatibility and simple postsynthesis purification processes. It is also necessary to develop synthesis methods that can easily apply various components to the surface of nanoflowers.

## 3. Biomedical Applications of Nanoflowers

Inorganic nanoflowers have demonstrated promising results in a variety of biomedical applications such as antibacterial treatment, biosensors, drug delivery, and combinational therapy. To the best of our knowledge, the scope of biomedical applications of organic nanoflowers is relatively very limited and the research is still in its early stages [42]. On the other hand, organic–inorganic nanoflowers have been widely applied in biomedical applications as biosensors to identify pathogens [101], cholesterol [102], dopamine [103], DNA [104], and micro RNA [105]; and as biomedicines such as drug and gene carriers [106,107] and for spinal cord injury treatment [108] and hemostasis [109]. Research on the biological application of inorganic nanoflowers has been actively attempted, but more diverse studies are still needed. This section will introduce several key studies. Several inorganic elements, such as Ag, Au, Pt, Si, Cu, CuO, and ZnO, have been applied as the main materials in the field of biomedical applications using nanoflowers due to their unique characteristics. Ag has been used in medicine for many years and is known as a potent antibacterial agent [110,111] Ag ions punch holes in bacterial membranes and create havoc once inside. Ag nanoscale materials have a greater inhibitory effect than bulk metallic forms or ionic forms [112]. For example, Ag nanoparticles have a higher antimicrobial activity against a wide range of bacteria, fungi, and viruses due to their high specific surface area and large surface-to-volume ratio [113,114]. In addition, nanoparticles are of particular interest due to their local surface plasmon resonance properties. These properties create other unique properties that are useful in such applications as antibacterial agents, chemical/biological sensors, biomedicine materials, SERS, and so on [115,116,117,118]. Au nanostructures have a high chemical stability, biocompatibility, plasmon tunability, and versatility in chemical modification [119,120]. Au nanoparticles showed potential in bioimaging and biosensing [121] and were proposed as therapeutic carriers for cancer treatment [122]. Pt exhibits high stability and is not easily oxidized [123]. In addition, Pt-based materials have been extensively studied due to Pt’s excellent catalytic ability in many applications such as organic catalysts [124], fuel cells [125], sensors [126], and cancer chemotherapy [127]. Pt nanostructures with a high electrocatalytic efficiency, sensitivity, and selectivity have been used in the manufacture of electrochemical sensors and biosensors [128,129,130]. Bulk Si is nontoxic, inexpensive, and the second most abundant element in the earth’s crust [131]. Si nanocrystals, which have advantages such as a low toxicity, high biocompatibility, and unique size and surface-dependent optical properties, have been utilized for bioimaging applications [132]. Noble metal nanoparticles, including Ag, Au, and Pt, have been intensively studied for biomedical applications; however, due to their high associated costs, various metals and metal oxides such as Cu, CuO, ZnO, and NiO have been studied as alternatives [76,133,134,135,136,137,138].

### 3.1. Antibacterial Treatment

The antibacterial properties of flower-shaped nanostructures bearing various elements have been extensively studied in the literature. Due to the small size and large surface-to-volume ratio of nanoflowers, they can directly interact with and disrupt membranes in biological systems with high efficiency [139]. Nanoflowers, which can improve antibacterial effects such as wound healing, have been used in the development of medical devices and implants. Ag nanoflowers synthesized using *Kalanchoe daigremontiana* extracts and CuO nanoflowered surfaces exhibited a high antibacterial activity against the Gram-negative bacteria *Escherichia coli* and the Gram-positive bacteria *Staphylococcus aureus* [16,100]. Perineum ZnO nanoflowers exhibited greater antibacterial activity against the Gram-positive bacteria *Staphylococcus aureus* than against the Gram-negative bacteria *Pseudomonas aeruginosa* [77]. The bactericidal rate of Si@NiOOH at 200 mg mL^−1^ was 99.9% against *Pseudomonas aeruginosa, Klebsiella pneumoniae,* and methicillin-resistant *Staphylococcus aureus*, whereas it exhibited negligible cytotoxicity toward mouse embryonic fibroblasts [24]. Notably, the morphology of Si@NiOOH was maintained even after its bactericidal activity [24]. Yan et al. engineered Au cores with AgAu shell alloy nanoflowers (Au@AgAu ANFs) [140]. Due to the rough surface morphology of the alloy, the Au@AgAu ANFs firmly adhered to bacteria and damaged their cell membranes. The ANFs showed highly stable (30 days) and long-lasting (48 h) antibacterial activity against *Escherichia coli* and remarkable biocompatibility with human neuroblastoma cells (SH–SY5Y) at a high concentration of 40 μg mL^−1^. The antibacterial efficacy of Au@AgAu ANFs was investigated in mouse intestine (Figure 8a). Four groups of the bacterially infected mice were treated with PBS (control group), Au@AgAu ANFs, Ag nanoparticles (Ag NPs), and kanamycin (Figure 8b,c). The antibacterial activity of the ANFs was similar to that of the kanamycin antibiotic in in vivo experiments; the ANFs demonstrated no cytotoxicity.

### 3.2. Biosensors

Biosensors are biologically derived from analytical devices that convert a biological response into an electrical signal [141,142,143,144,145]; they are typically composed of transducers for biological recognition units and signal-converting systems. Researchers have developed highly selective and sensitive biosensors for a wide range of applications such as disease diagnosis and monitoring food quality.

#### 3.2.1. Glucose Monitoring

Diabetes mellitus is a metabolism disorder that elevates blood sugar (glucose) levels to ≥6.9 mM on an empty stomach; it can lead to death and disability [146]. Therefore, developing sensors to monitor glucose has attracted considerable attention from researchers worldwide.

For the diagnosis and management of diabetes mellitus, nonenzymatic glucose sensors were developed by fabricating Pt nanoflowers on Au electrodes via a template-free ultrasonic electrodeposition method [147]. The differential pulse voltammograms showed that the Pt nanoflower electrodes for glucose determination exhibited a sensitivity of 2217 μA mM^−1^ cm^−2^ (+0.3 V), a linear calibration range of 1–16 mM, and a detection limit of 48 μM at a signal-to-noise ratio (S/N) of 3.

Ag nanoflowers fabricated via the cyclic scanning electrodeposition method exhibited excellent electrocatalytic activity with a low detection limit of 0.1 nM (S/N = 3) and a high sensitivity of ~4230 mA cm^−2^ mM^−1^ [148,149]. The electrocatalytic activity was attributed to the curved nanopetals with a high density of atomic steps [150], effective area of the Ag(OH)_ad_ layer, good electron transport by the continuous 3D intercrossed Ag petals due to their high surface-to-volume ratios, and excellent interfacial contact between the flowerlike Ag nanoparticles and substrate via the bridge linker 3-mercaptopropyltrimethoxysilane [151].

Kong et al. synthesized hierarchical CuO nanoflowers using Cu_2_O nanocubes as templates. The CuO nanoflower-modified electrodes exhibited a higher sensitivity (2217 μA mM^−1^ cm^−2^), lower detection limit (0.96 μM), and broader linear range (up to 6 mM) for nonenzymatic glucose sensing than those of the other reported sensors (Table 1) [92]. These electrodes demonstrated a fast response time, long-term stability, and good practical applicability in determining glucose levels in human blood serum samples.

**Table 1 pharmaceutics-14-01887-t001:** Comparison of the performance of the CuO-nanoflower-modified electrode with those of the other reported glucose sensors. Reprinted with permission from Ref. [92]. Copyright 2018, Elsevier.

Electrode Materials	Sensitivity (μA·mM^−e^·cm^−c^)	Linear Range (up to mM)	Detection Limit (μM)	References
CuO/Cu_2_O/Cu	1541	4	0.57	[152]
CuO/Au	1101	13.3	50	[153]
Nanoporous CuO/Cu	1066	2.04	~	[154]
CuO nanowire/Cu	1420.3	2.05	5.1	[155]
CuO/Gox	47.19	10.0	1.37	[156]
CuO nanoflowers	2217	6	0.96	[92]

#### 3.2.2. Hydrogen Peroxide (H_2_O_2_) Sensors

H_2_O_2_ is a strong oxidizing agent that is a catalytic byproduct of oxidases such as glucose oxidase, cholesterol oxidase, and lactate oxidase [157] and a precursor in the formation of hydroxyl radicals [158]. Therefore, the development of a sensitive, convenient, and fast H_2_O_2_ sensor is highly desirable for disease diagnosis. The hierarchical, porous CuO/Cu nanoflower-modified electrode materials for nonenzymatic H_2_O_2_ sensors were synthesized via surfactant-free oxidation of a Cu powder in alkaline solution [17]. These nanoflowers were characterized using cyclic voltammetry and amperometry under alkaline conditions; they exhibited a high sensitivity (103 μA mM^−1^ cm^−2^), low detection limit (2 μM/L), and broad concentration range (2 μmol L^−1^–19.4 mmol L^−1^). To determine the effect of the oxidation of human serum on the amperometric response of H_2_O_2_, the current-time responses at the CuO/Cu/glassy carbon electrode with added disruptors such as H_2_O_2_, uric acid, ascorbic acid, and l-cysteine were investigated [159]. The effects of these interferants were negligible, indicating that the CuO/C nanoflowers demonstrated good selectivity for H_2_O_2_ detection. The long-term stability of the CuO/Cu nanoflower-modified electrode was studied by measuring the current response to H_2_O_2_ for 30 days under alkaline conditions and was observed to be 88.4%. These excellent features (stability, sensitivity, anti-interference property, and wide concentration range) of the nonenzymatic H_2_O_2_ sensor with the CuO/C nanoflower-modified electrode were attributed to the large specific surface area and porosity of the nanoflowers, stable nanostructure, and enzyme-free detection.

#### 3.2.3. H_2_O_2_ and Glucose Dual Sensors

CuO nanoflowers were fabricated on a glassy carbon (GC) electrode as a dual-function amperometric sensor for H_2_O_2_ and glucose [18]. The CuO nanoflowers/GC electrode exhibited an excellent response to H_2_O_2_ with a high sensitivity (956.69 μA mM^−1^ cm^−2^) and wide linear range (0.005–14.07 mM). In addition, the experiment showed a high electrocatalytic activity for glucose oxidation with a high sensitivity (1086.34 μA mM^−1^ cm^−2^) and low detection limit (0.12 μM, S/N = 3) [18].

#### 3.2.4. Amyloid Detection

Amyloids are misfolded protein aggregates that have been linked to amyloidosis and neurodegenerative diseases such as Alzheimer’s disease [160,161], Parkinson’s disease [162], prion-related diseases [163], and type 2 diabetes [164]. A rapid, cost-effective, and sensitive ZnO-nanoflower-based nano-biosensor was developed for amyloid detection [165]. Interestingly, ZnO nanoflowers and nanoparticles were also used for amyloid degradation [19]. Fluorescence studies with Thioflavin T, atomic force microscopy, infrared spectroscopy, and fibril size reduction using dynamic light scattering on a model human insulin amyloid indicated that ZnO nanoflowers had a higher anti-amyloid ability than that of ZnO nanoparticles due to the higher surface-to-volume ratio of the nanopetals [19].

### 3.3. Drug Delivery and Combinatorial Treatment

Due to their structural features and physical properties, nanoflowers are suitable for combinational therapies that include drug delivery. Li et al. synthesized H-AuNFs and evaluated their drug-loading capacity and pH/near-infrared (NIR) controlled-release properties using doxorubicin hydrochloride (DOX) as a model anticancer agent for synergistic chemo-photothermal cancer therapy [20]. The in vivo tumoricidal efficacy was investigated in tumor-bearing Kunming mice (Figure 9a). A histological examination of lung, liver, spleen, kidney, and heart was performed to monitor the toxicity of H–AuNFs in the treated mice (Figure 9b). When the nanoflowers were synthesized from a plasmonic metal element such as Au, they exhibited long-wavelength LSPR due to the protruding petals. Moreover, the local temperature could be elevated through NIR laser irradiation and photothermal conversion effect [166,167,168]. These results indicated that H–AuNFs exhibit excellent biocompatibility, high photothermal conversion efficiency (ƞ = 52%), pH/NIR dual-responsive drug delivery, and synergistic chemo-photothermal efficacy.

In summary, inorganic nanoflowers have shown promising results in biomedical applications such as antibacterial treatment; biosensors to detect glucose, H_2_O_2_, and amyloids; drug delivery; and combinatorial therapy. Compared to the research results for hybrid nanoflowers or other nanomaterials (e.g., gelatin nanofibrous scaffolds for engineering cardiac tissues [169]; Ag nanoparticles for adhesives, wound closing, and hemostatic [170,171,172]; and hydrogel nanoparticles for drug release and delivery [173]), the application area is relatively limited and the biomedical application research of inorganic flowerlike nanostructures is still in its early stages.

## 4. Summary

Herein, we presented a comprehensive review of the recent advances in inorganic-element-based nanoflowers used in biomedical applications. The synthetic strategies for nanoflowers that had biocompatibility, improved efficiency, and specific functions and structures for biomedical applications were described. An eco- and biofriendly synthesis method must be developed by minimizing the amounts of toxic residues and eliminating seeds, surfactants, and templates as much as possible. The synthetic strategies of hollow-structured nanoflowers and wrapping porous flowerlike shells around core nanoparticles were designed for specific bioapplications, drug delivery, and combinational therapy.

These nanoflowers can be used for their antibacterial effects such as wound healing; for manufacturing medical devices and implants as well as biosensors to detect diseases such as diabetes, food poisoning, amyloidosis, and neurodegenerative diseases; and for facilitating drug delivery.

Synthetic methods for inorganic flower-shaped nanoparticles consisting of metals and metal oxides and their particle size distribution and applications are summarized in Table 2.

**Table 2 pharmaceutics-14-01887-t002:** Inorganic hierarchical flowerlike nanomaterials and their biomedical applications.

Nanostructures	Production Methods	Size	Application	Refs.
Au nanoflowers	Vesicle-directed generation	406 ± 89 nm	SERS	[33]
Au nanoflowers	Seed-mediated method	55 nm	SERS-mapping immunoassay	[174]
Hollow-channel Au and Ag nanoflowers	Template method	193 ± 47 nm	Catalysts and SERS	[175]
Au nanoflowers, nanostars, and nanosnowflakes	Seedless and surfactant-free approach	100 nm (nanoflowers) 60–70 nm (nanostars) 90 nm (nanosnowflakes)	Catalyst and photothermal therapy	[121]
Pt nanoflowers	Template-free synthesis	–	Surface-assisted laser desorption/ionization mass spectrometry analysis of biomolecules	[124]
Pt nanoflowers	Sonoelectrodeposition method	–	Electrocatalysts and nonenzymatic sensors	[176]
Branched Ag nanoflowers	Biosynthesis	40–60 nm	SERS and antibacterial treatment	[177]
Ag nanoflower	Cyclic scanning electrodeposition method	~5.5 μm	Catalysts of nonenzymatic electrochemical glucose biosensors	[148]
Multibranched AgPt alloyed dendritic nanoflowers	One-pot successive coreduction aqueous method	332.7 nm	SERS	[178]
CuO nanoflowers	Hydrothermal method	1 μm	H_2_O_2_ sensor	[179]
Flower-shaped CuO nanostructures	Biosynthesis	~250 nm	Photocatalysts and antibacterial agents	[76]
CuO nanospindles and CuO nanoflowers	Green synthesis	–	Antimicrobial agents	[75]
ZnO nanoflowers	Hydrothermal method	–	Antibacterial agents	[180]
ZnO nanoflowers	Solution method	316 nm	Antiamyloid agents	[19]
ZnO nanoflowers	Hydrothermal method	–	Anticancer agents	[181]
ZnO/Ag nanoflowers	Hydrothermal method	1.5–3.5 μm	SERS	[182]
Fe_3_O_4_ nanoflowers	Solvothermal route	70–80 nm	Theranostic applications, such in for phototherapy and magnetic resonance imaging	[183]
Fe_0.6_ Mn_0.4_O nanoflowers	Thermal-decomposition reaction	102.7 ± 11 nm	Diagnostic applications and therapeutic interventions through magnetic hyperthermia	[184]
γFe_2_O_3_@Au core–shell-type nanoflowers	Coprecipitation method [185] and iterative growth	179 nm	Theranostic applications	[186]

## 5. Future Perspectives

Future research must focus on the development of 3D flowerlike nanomaterials with a uniform size and shape as well as improved performance via existing ecofriendly synthesis methods. As the development of inorganic-element-based nanoflowers for biomedical applications is limited thus far, nanoflowers consisting of various inorganic components or composite structures must be developed.

Nanoflowers have potential applications in other bioscience fields such as biomaterials, medicine, and biotechnology. However, further research should focus on their applicability and improvements in the design of structured or composite nanoflowers with significantly attractive properties.

## Figures and Tables

**Figure 1 pharmaceutics-14-01887-f001:**
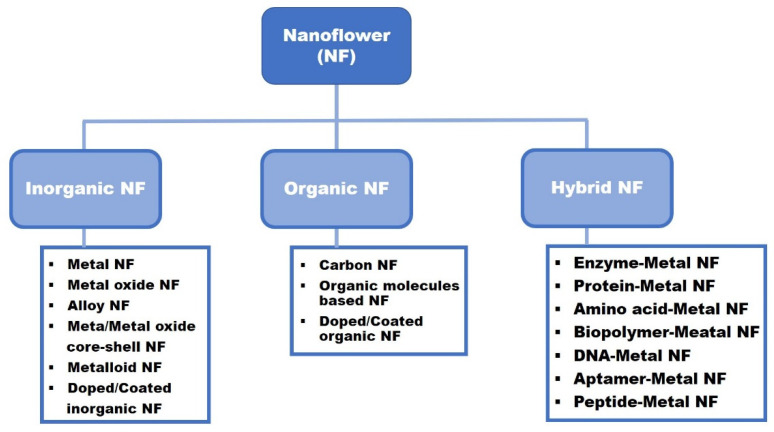
Nanoflower classification schematic according to the composition.

**Figure 2 pharmaceutics-14-01887-f002:**
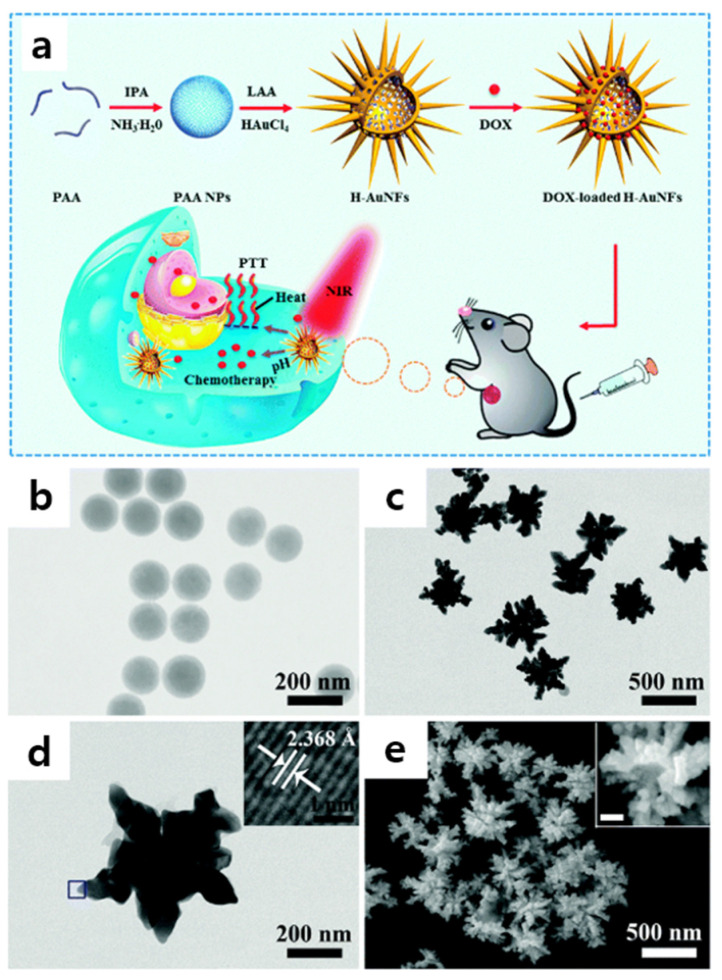
(**a**) Schematic of the synthetic strategy for H–AuNFs as pH/near-infrared (NIR) dual-responsive drug vehicles for in vitro and in vivo synergistic chemo-photothermal cancer therapy. IPA = isopropyl alcohol, LAA = l-ascorbic acid, DOX = doxorubicin hydrochloride. Transmission electron microscopy (TEM) images of (**b**) polyacrylic acid nanospheres and (**c**) H–AuNFs. (**d**) High-resolution transmission electron microscopy (HR-TEM) image of a single H–AuNF. Inset: magnification of the area marked with a square. (**e**) Scanning electron microscopy (SEM) image of H–AuNFs. Inset: SEM image of a broken H–AuNF (scale bar: 100 nm). Reprinted with permission from Ref. [20]. Copyright 2015, The Royal Society of Chemistry.

**Figure 3 pharmaceutics-14-01887-f003:**
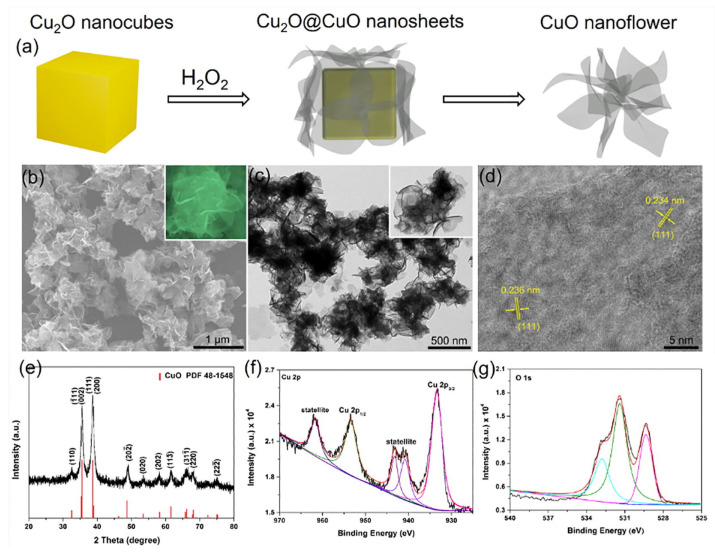
(**a**) Schematic of the template-based synthesis of CuO nanoflowers; (**b**) low-magnification field emission scanning electron microscopy (FE-SEM), (**c**) TEM, and (**d**) HR-TEM images; (**e**) XRD pattern; and (**f**) Cu 2p and (**g**) O 1s XPS spectra of CuO nanoflowers. Insets in (**b**,**c**) are the corresponding individual nanoflowers. Reprinted with permission from Ref. [92]. Copyright 2018, Elsevier.

**Figure 4 pharmaceutics-14-01887-f004:**
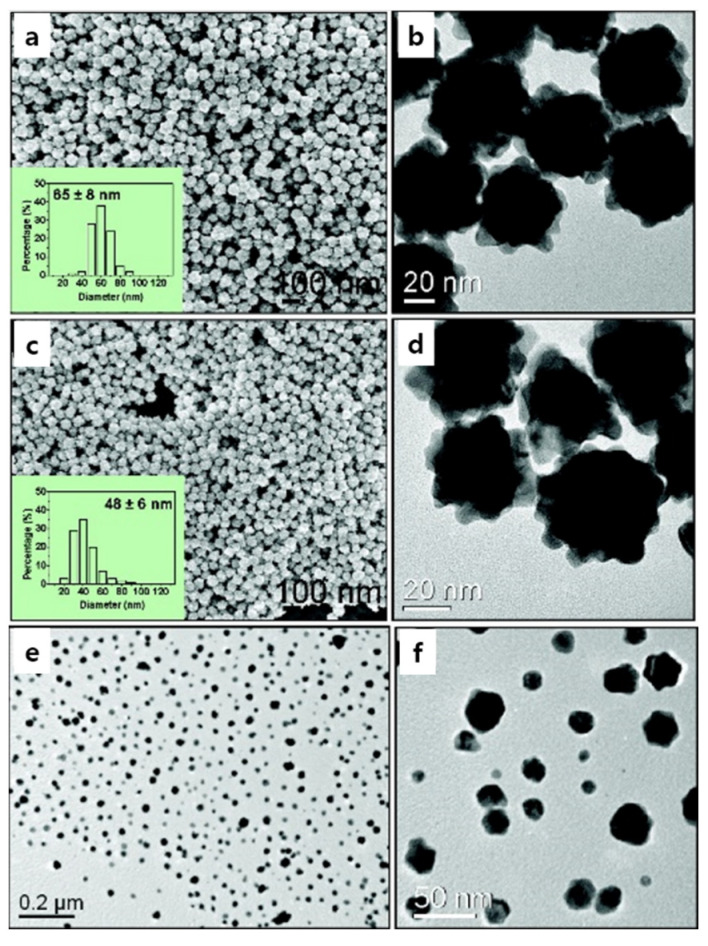
Representative FE–SEM and TEM images of Au nanocrystals formed by reducing aqueous AuCl_4_^−^ solution (0.5 mM) with HEPES solutions of different concentrations: (**a**,**b**) 15 mM; (**c**,**d**) 20 mM; (**e**,**f**) 40 mM. The insets in (**a**,**c**) show the histograms of the size distribution of the as-synthesized Au nanoflowers. Reprinted with permission from Ref. [52]. Copyright 2008, American Chemical Society.

**Figure 5 pharmaceutics-14-01887-f005:**
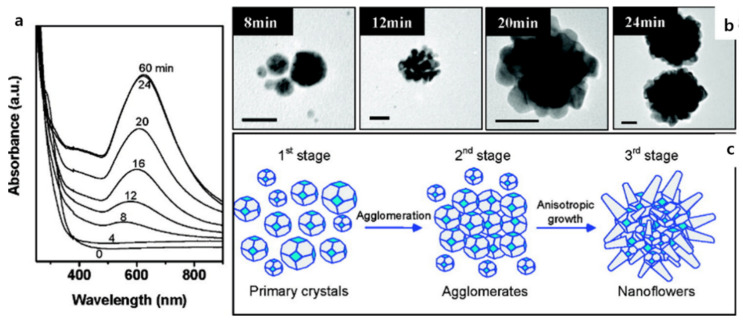
(**a**) UV–vis spectra as a function of reaction time of aqueous AuCl_4_^−^ solution (0.5 mM) and HEPES (10 mM). (**b**) Representative TEM images of the products harvested after 8, 12, 20, and 24 min of reaction time. All scale bars are 20 nm. (**c**) Schematic of the proposed mechanism for the formation of Au nanoflowers in HEPES solution. Reprinted with permission from Ref. [52]. Copyright 2008, American Chemical Society.

**Figure 6 pharmaceutics-14-01887-f006:**
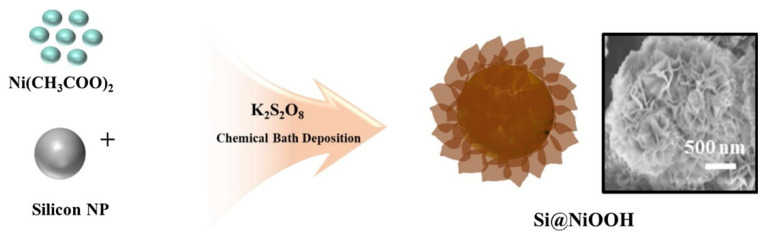
Schematic of Si@NiOOH prepared from Si particles and Ni(II) acetate. Reprinted with permission from Ref. [24]. Copyright 2021, Elsevier.

**Figure 7 pharmaceutics-14-01887-f007:**
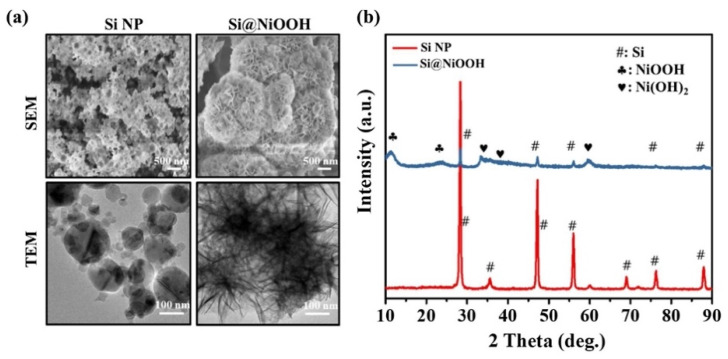
(**a**) SEM (**top**) and TEM (**bottom**) images of Si nanoparticles (SiNP) (**left**) and Si@NiOOH (**right**). (**b**) XRD of Si NP (red) and Si@NiOOH (blue). Scale bars = 500 nm and 100 nm, respectively. Reprinted with permission from Ref. [24]. Copyright 2021, Elsevier.

**Figure 8 pharmaceutics-14-01887-f008:**
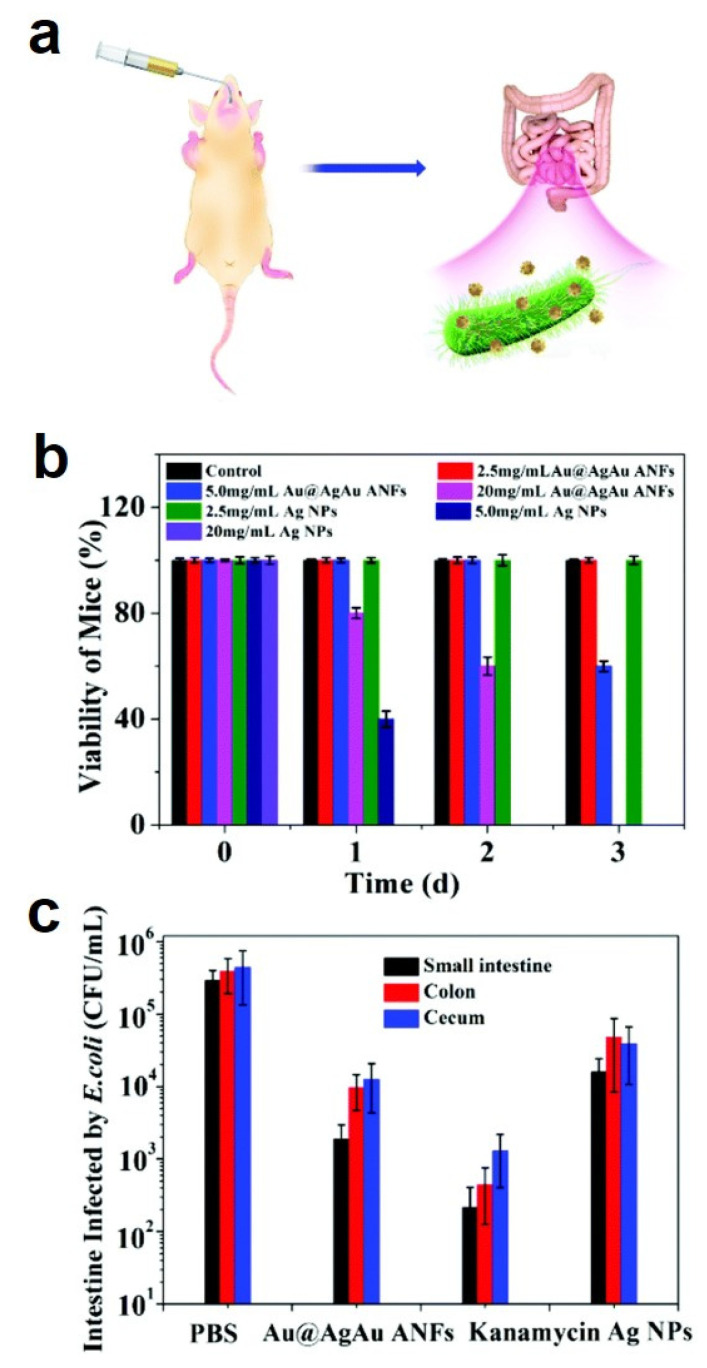
(**a**) Schematic of the evaluation of the in vivo antibacterial activity of Au@AgAu alloy nanoflowers (ANFs). (**b**) Viability of the mice treated with different concentrations of Ag NPs and Au@AgAu ANFs for three days. (**c**) Surviving *E. coli* in the small intestine, cecum, and colon on the fourth day after treatment with PBS, Au@AgAu ANFs, kanamycin, and Ag NPs. Reprinted with permission from Ref. [140]. Copyright 2018, The Royal Society of Chemistry.

**Figure 9 pharmaceutics-14-01887-f009:**
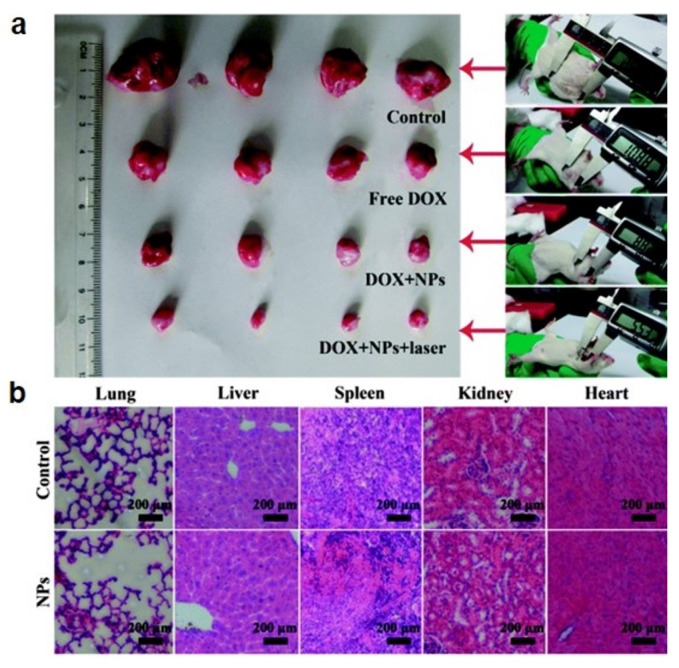
(**a**) Representative photographs of excised tumors from each group on the 11th day after treatment (**left**) and images of Kunming mice with tumors (**right**). (**b**) Hematoxylin-and-eosin-stained histological sections of major organs (heart, liver, spleen, lung, and kidneys) from mice treated with the control (saline) and H–AuNFs. Reprinted with permission from Ref. [20]. Copyright 2015, The Royal Society of Chemistry.

## Data Availability

Not applicable.

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
