# Peer review of "Inorganic Nanoflowers—Synthetic Strategies and Physicochemical Properties for Biomedical Applications: A Review"

_pharmaceutics, 2022, doi:10.3390/pharmaceutics14091887_

Round 1

Reviewer 1 Report

The submitted manuscript by Lee et al. is a review article about inorganic nanoflowers. This review is a very interesting manuscript and, overall, well written. The authors have discussed the synthesis, properties and recent advances in the biomedical applications of inorganic nanoflowers. However, this review paper is about inorganic nanoflowers; the authors should have included a brief introduction to organic and hybrid nanoflowers. A comparison between all nanoflower structures, organic, inorganic and hybrid, should be made, giving the reader a comprehensive idea about nanoflowers.

My comments are given below.

    - You should include a brief section about organic and hybrid nanoflowers highlighting their synthetic routes, properties and applications. [I will suggest the section on organic nanoflowers be incorporated at the begging of the central part, while the section about hybrid nanoflowers should be at the end of the central part]

-     -  Comparing organic, inorganic and hybrid structures will enhance the need for further study of inorganic nanoflowers.

       -  It will be easier for the reader to start the discussion for inorganic nanoflowers with the synthesis, not the applications. The applications are directed by the synthetic route most of the time. Thus, the reader needs to have an idea first.

-      -  About the synthesis of inorganic nanoflowers, you could discuss the selection of elements used for synthesis. 

Author Response

Reviewer 1

The submitted manuscript by Lee et al. is a review article about inorganic nanoflowers. This review is a very interesting manuscript and, overall, well written. The authors have discussed the synthesis, properties and recent advances in the biomedical applications of inorganic nanoflowers. However, this review paper is about inorganic nanoflowers; the authors should have included a brief introduction to organic and hybrid nanoflowers. A comparison between all nanoflower structures, organic, inorganic and hybrid, should be made, giving the reader a comprehensive idea about nanoflowers.

My comments are given below.

    - You should include a brief section about organic and hybrid nanoflowers highlighting their synthetic routes, properties and applications. [I will suggest the section on organic nanoflowers be incorporated at the begging of the central part, while the section about hybrid nanoflowers should be at the end of the central part]

Response: Thank you for your valuable comment, we added a brief section about organic and hybrid nanoflowers to highlight their synthetic routes, properties and applications into the middle section of introduction, the front paragraph of synthesis and characterization and application as following as

Organic nanoflowers are composed of organic molecules only, or inorganic elements are present as part of the medium in which organic molecules are the main components [35-39]. For instance, the carbon nanoflowers synthesized using electric arc discharge method in water are reported and the obtained carbon nanoflowers were composed by highly crystalline graphene nanosheets, which is arranged like flowers [40]. Zheng’s group introduced nitrogen, phosphorous, and fluorine doped carbon nanoflowers by ultrasound-induced polycondensation and pyrolysis [41]. Also, organic nanoflowers are constructed from a variety of molecules (guests) using fixed supramolecular hacky sacks, which are hierarchically structured particles, as templates [42]. The nanoflowers with spiky and wide petals are produced using small/rigid molecules (e.g., doxorubicin) and large/flexible biomacromolecules (e.g., proteins and plasmid DNA), respectively, as guests [42].

Organic-inorganic hybrid nanoflower, also called hybrid nanoflowers, are defined as all component of inorganic nanostructures being associated with organic materials [43]. Generally, organic components include enzymes, protein, amino acids, biopolymer, DNA, and peptides, containing amide or amine groups to form complex with metal ions by coordination interaction. Inorganic materials are mostly composed of divalent metals such as Cu2+, Zn2+, Ca2+, Fe2+ and Mn2+ [44]. Ge and co-workers [45] discovered the first hybrid nanoflowers, confirming that Cu2+ ions and proteins can make novel typed nanoparticles. The four types of hybrid nanoflowers are prepared using α-lactalbumin, laccase, carbonic anhydrase, and lipase. The formation of dual enzyme inorganic hybrid nanoflowers are reported using glucose oxidase and lipase as the organic materials, and Cu3(PO4)2·3H2O as the inorganic components [46]. Li et al. synthesized carbon nanotube embedded lipase-Ca/Fe/Cu nanoflowers as a biocatalysts of the chiral resolution reaction [47].

   Organic nanoflowers were synthesized in similar synthesis technologies to those of inorganic nanoflowers. The electric arc discharge method [40], ultrasound-induced polycondensation and pyrolysis [41], reduction–pyrolysis–catalysis route [83], chemical vapor deposition [84], and microwave assisted hydrothermal-hydrothermal-carbonization-etching method [85,86] were reported.

In case of the organic-inorganic hybrid nanoflower firstly reported, in 2012, the mild and direct co-precipitation method was developed to synthesize hybrid protein–copper phosphate nanoflowers [45]. Since then, various hybrid nanoflowers have been synthesized based on the co-precipitation method, which was carried out by mixing the organic elements (e.g., enzymes, protein, amino acids and so on) and metal ions: Cu2+, Zn2+, Mn2+, Fe2+, and Co2+, in the presence of phosphate buffered saline or directly using metal phosphate. And then, the mixture was incubated or sonicated [87-91].

To the best of our knowledge, the scope of application of organic nanoflowers is relatively very limited, and the research is still in its early stages [101]. On the other hand, organic-inorganic nanoflowers are widely applied in biomedical applications as biosensors to identify pathogens [102], cholesterol [103], dopamine [104], DNA [105], and micro RNA [106], biomedicines such as drug and gene carriers [107,108] and spinal cord injury treatment [109], and hemostasis [110].

- Comparing organic, inorganic and hybrid structures will enhance the need for further study of inorganic nanoflowers.

Response: Thank you for your kind comment. We briefly introduce organic, inorganic and hybrid nanoflowers and compared their composition, preparation methods and structures following your comments.

     - It will be easier for the reader to start the discussion for inorganic nanoflowers with the synthesis, not the applications. The applications are directed by the synthetic route most of the time. Thus, the reader needs to have an idea first.

Response: Thank you, we changed the discussion in order of followings, 1. Introduction; 2. Synthesis and characterization of nanoflowers; 3. Biomedical applications of nanoflowers; 4. Summary; 5. Future perspective.

 -  About the synthesis of inorganic nanoflowers, you could discuss the selection of elements used for synthesis. 

Response: Thank you for your valuable comment. We discussed the selection of elements used for biomedical application rather than for synthesis on the section of biomedical applications of nanoflowers as following as 

Several inorganic elements have been applied as their main materials in the field of biomedical applications using nanoflowers, such as Ag, Au, Pt, Si, Cu, CuO, and ZnO, due to their unique characteristics. Ag has been used in medicine for many years and known as a potent antibacterial agent [111,112] Ag ions punch holes in bacterial membranes and cause havoc once inside. Ag nanoscale materials have a greater inhibitory effect than bulk metallic forms or ionic forms [113]. For example, Ag nanoparticles have higher antimicrobial activity, due to their high specific surface area and the large surface to volume ratio against a wide range of bacteria, fungi, and viruses [114,115]. In addition, the nanoparticles are of particular interest due to their local surface plasmon resonance properties. These properties give unique properties that are useful in areas such as antibacterial agents, chemical/biological sensors, biomedicine materials, SERS and so on 116-119]. Au nanostructures have high chemical stability, biocompatibility, plasmon tunability, and versatility in chemical modification [120,121]. Au nanoparticles showed potential in bioimaging and biosensing [122] and were proposed as therapeutic carriers for cancer treatment [123]. Pt exhibits high stability and is not easily oxidized [124]. In addition, Pt-based materials have been extensively studied due to Pt’s excellent catalytic ability in many applications such as organic catalysts [125], fuel cells [126], sensors [127], and cancer chemotherapy [128]. Pt nanostructures with high electrocatalytic efficiency, sensitivity and selectivity have been used in the manufacture of electrochemical sensors and biosensors [129-131]. Bulk-Si is nontoxic, cheap, and the earth-abundant element in the crust [132]. Si nanocrystals have advantages such as low toxicity, high biocompatibility, and unique size and surface-dependent optical properties and are utilized for bioimaging applications [133]. Noble metal nanoparticles, including Ag, Au, and Pt, have been intensively studied for biomedical applications; however, due to their high associated costs, various metal and metal oxides such as Cu, CuO, ZnO, and NiO have been studied as alternatives [76,134-139].

Reviewer 2 Report

1.     The manuscript entitled ‘’Inorganic Nanoflowers-synthetic strategies and physicochemical properties for biomedical applications: A reviewcontains some interesting findings. However, the submitted manuscript requires significant improvement before I can recommend publication. This is due to an unclear scope of the manuscript, a mystifying structure, and the need for language proofreading. The manuscript has the problem of simply listing the literature, without further discussion and summary, just like your reading notes. It is recommended to add concluding observations in each paragraph/subsection. In the following, I will try to make concrete suggestions on how to improve this article.

2.       Insert some infographics to make the manuscript interesting to the reviewers and readers. Also, able to secure future citations and visibility.

3.       The introduction section is very short and should be improved entirely so that the reader can identify the scientific problems solved by this research. For instance, several biomaterials are used for biomedical applications by core researchers such as Nasim Annabi, Seeram Ramakrishna, Narsimha Mamidi, and others. so, it is recommended to cover/include such works in the revised manuscript.  

4.       It is recommended to use the keywords for literature research " nanoflowers, biomedical applications, drug delivery, wound healing, antibacterial treatment, combinatorial cancer therapy, amyloidosis, H2O2, biosensors, etc., or combinations of those words must be needed for this systematic review. So, more keywords should be included to make a literature search and an infographic of literature should be added to the revised manuscript.

5.       It is recommended to increase the description of the main synthesis methods of nanoflowers, and the influence of different synthesis methods on the properties of the nanoflowers in biomedical applications.

6.       A summarized table of clinical trials or inorganic nanoflowers approved by the FDA or WHO is recommended.

7.       The manuscript contained around 96 references, so more references should be added to the revised manuscript.

8.       A future perspective section should be added separately.

Round 2

Reviewer 2 Report

I don´t have any comments.